# Investigating the Potassium Fertilization Effect on Morphological and Agrophysiological Indicators of Durum Wheat under Mediterranean Rain-Fed Conditions

Amina Messaoudi [1,*], Fatiha Labdelli [1], Nazih Y. Rebouh [2], Malika Djerbaoui [1], Dmitry E. Kucher [2], Salah Hadjout [3], Walid Ouaret [4], Olga A. Zakharova [5] and Mourad Latati [6,*]

[1] Agro-Biotechnology and Nutrition Laboratory in Semi-Arid Zones, Ibn Khaldoun University, Tiaret 14000, Algeria; labdellifa@yahoo.fr (F.L.); kmsoilaz@yahoo.fr (M.D.)

[2] Department of Environmental Management (RUDN University), 6 Miklukho-Maklaya St, Moscow 117198, Russia; n.yacer16@outlook.fr (N.Y.R.); kucher-de@rudn.ru (D.E.K.)

[3] Centre de Recherche en Aménagement du Territoire (CRAT), Campus Zouaghi Slimane, Route de Ain el Bey, Constantine 25000, Algeria; hadjout.salah@gmail.com

[4] Ecology, Evolution, and Environmental Biology NASA Interdisciplinary Research in Earth Science Geospatial Analysis Center, Miami University, Oxford, OH 45056, USA; ouaretw@miamioh.edu

[5] Department of agronomy and agrotechnologies, Ryazan State Agrotechnological University Named after P.A. Kostychev, 1, Kostychev Str., Ryazan 390044, Russia

[6] Ecole Nationale Supérieure Agronomique (ES1603), Laboratoire d'Amélioration Intégrative des Productions Végétales (C2711100), Département de Productions Végétales, Avenue HassaneBadi, El Harrach, Algiers 16200, Algeria

* Correspondence: a.messaoudi.ensa94@gmail.com (A.M.); m.latati@yahoo.com (M.L.); Tel.: +21-367-160-6269 (M.L)

**Abstract:** It is well known that balanced and optimal mineral fertilization (i.e., nitrogen, phosphorus, and potassium) can increase yield and improve wheat quality. However, there is little scientific knowledge on the specific effect of potassium (K) fertilization on the morphological and qualitative characteristics of rainfed durum wheat under Mediterranean conditions, especially in the context of Algerian agriculture. Therefore, the main objective of this study is to provide crucial information on this important type of durum wheat mineral nutrition for farmers and researchers working in similar areas. The field study was conducted in the Northern Algerian region of Mitidja during the 2020/2021 and 2021/2022 growing seasons. The effect of K fertilization was evaluated within five contrasted rates of K: 0 kg $k_2O$ ha$^{-1}$, 100 kg $k_2O$ ha$^{-1}$, 200 kg $k_2O$ ha$^{-1}$, 300 kg $k_2O$ ha$^{-1}$, and 400 kg $k_2O$ ha$^{-1}$, which were applied on one of the most commonly cultivated cultivars of durum wheat (Simeto). Results showed that increasing potassium levels had a positive and significant impact only on grain yield, spike length, spike neck, and dry matter. Hence, the highest grain yield of durum wheat (2.58 t ha$^{-1}$) was demonstrated under low K application (100 kg $k_2O$ ha$^{-1}$). This rate leads to an increase in grain yield by more than 0.6 t ha$^{-1}$, as compared to the unfertilized control. Moreover, the thousand-grain weight of durum wheat was significantly increased by 4.11 g and 1.96 g, respectively, under low and moderate (200 kg $k_2O$ ha$^{-1}$) K-fertilization, as compared to the control. In terms of grain yield quality, low K application provides an increase of 1.21% in protein content as compared to that measured under the control treatment. The major finding confirmed that both growth and yield indicators of durum wheat were globally optimized under low K application. Based on grain yield and evaluated agronomic traits, this research revealed that an applied K rate of 100 kg $k_2O$ ha$^{-1}$ is recommended as the most effective dose to maximize durum wheat yield and quality under Algerian sub-humid conditions.

**Keywords:** durum wheat; morphological traits; qualitative characteristics; potassium fertilization; Algerian sub-humid conditions





## 1. Introduction

Durum wheat (*Triticum durum*) is an essential strategic crop for food security because it is a main source of protein, which provides gluten for bread making [1]. In Algeria, durum wheat is consumed in various forms: paste, bread, and couscous [2]. At the national level, it is the most cultivated cereal, and its current yield is around 1800 kg ha$^{-1}$ [3]. Previous studies have shown that a yield of 4000 kg ha$^{-1}$ would be necessary to meet the needs of the population. However, several factors, such as water deficit, seed quality, sowing date, and inadequate use of fertilizers, influence wheat crop productivity [4,5]. Among these factors, drought stress is a very important abiotic stress that constricts wheat production [6,7]. According to a report published by the United States Department of Agriculture (USDA), the water deficit has caused a decline in wheat production, which has led to imports of about 8 million tons for the year 2021 [8]. Wherefore, rain-fed cropping systems should be considered as one of the major ways for increasing and stabilizing cereals production, especially in sub-humid Algerian areas [9]. The cultivated area by durum wheat covered an average of 22,496 ha, in which durum wheat is generally cropped in rotation with either legumes or farmed through fallow practice [10–12].

Potassium is the most abundant cation in plants [13]. In the soil, potassium is found in four forms: exchangeable, constitutive, retrograde, and soluble [14]. The exchangeable and soluble forms constitute the compartment available to plants. The potassium-supplying capacity of soil depends on its total potassium content and the releasing characteristics of different forms, which are influenced by the physico-chemical properties of the soil [15]. Its uptake by the plant decreases with decreasing soil water content [16]. Indeed, it has multiple functions that affect the physiological functioning of plants, including improving plant quality, the photosynthetic process, osmotic regulation, enzyme activity, assimilate stimulation and transport, protein synthesis, and stress tolerance [17,18]. Nitrogen and phosphate fertilizers are the main fertilizers used in wheat production, while potassium fertilizer is rarely applied [19], and its deficiency reduces the qualitative and quantitative yield of wheat crops [20]. Additionally, in the absence of adequate potassium fertilization, significant soil K deficiency occurs [15]. Halillat [21] and Belaid [22] have shown that potassium fertilization has a favorable effect on agronomic behavior, including certain characteristics of durum wheat, such as yield and thousand-grain weight.

Despite the importance of the role of potassium in the formation of yield, the practice of potassium fertilization on the wheat crop remains very limited in Algeria [23]. The main objective of the present study is to identify the optimal K fertilizer dose that could stabilize and improve growth and yield components of durum wheat crop, particularly under organic cropping management in sub-humid climate. We hypothesize that both low and moderate K application will maintain simultaneously yield and K acquisition by durum wheat. The specific objectives of this study were (i) to evaluate the influence of different K fertilizer application rates on wheat yield, (ii) to understand the response of the crop to variation in K use on morphological traits with different rates of potash fertilizer, and (iii) to determine the influence of potassium rates on the quality of wheat grain. The field experiment was conducted within the same conditions of crop and soil management that are commonly practiced by the local farmers under rain-fed conditions.

## 2. Materials and Methods

### 2.1. Study Area Characteristics

The field experiment concerns the evaluation effect of different levels of potassium fertilizer on the growth and yield of durum wheat under rainfed conditions, during two successive growing seasons: 2020–2021 and 2021–2022. The experiment is set in the North-East of the Mitidja plain (Zone 1 in Figure 1) at the Technical Institute of Field Crops in Oued Semar Algiers; it is a pilot farm for the production of cereals and also the multiplication of the seeds. This station extends over an area of approximately 100 ha (latitude 36°43′ north, longitude 30°08′, altitude 24 m) on a type of soil «fluvisol». This soil type covers the whole surface of this station and also a large part of the surface of the Mitidja plain.

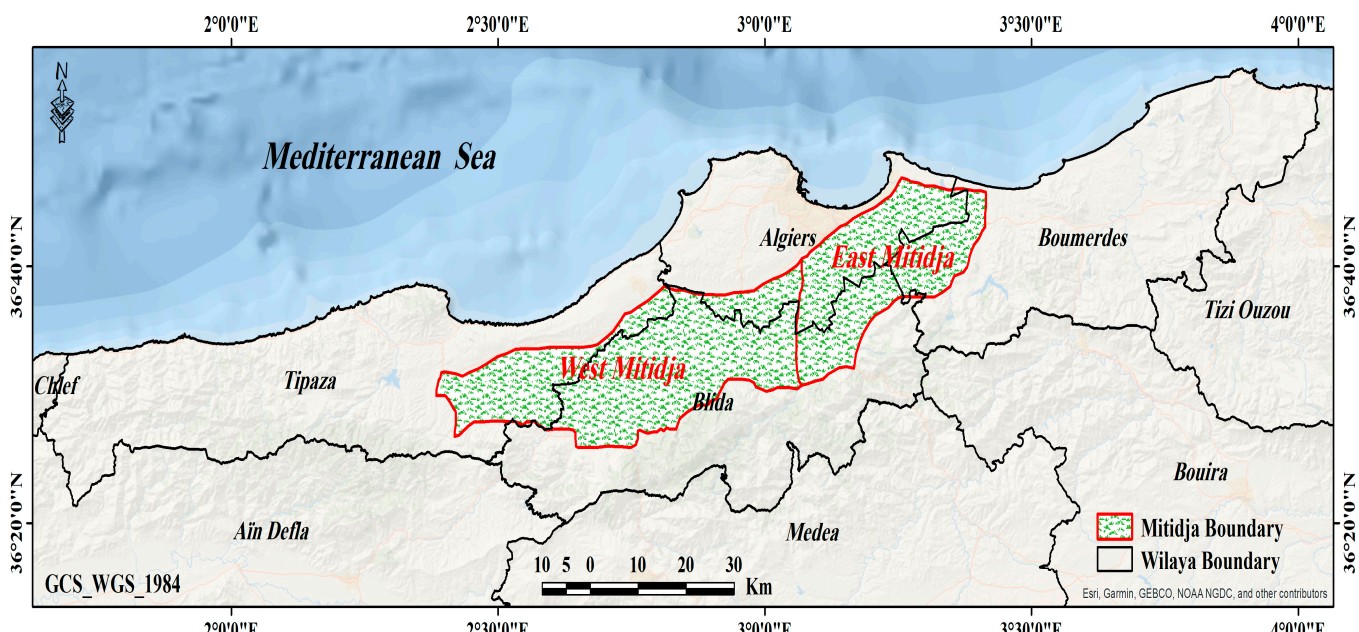

**Figure 1.** The extent of the Mitidja plain in northern Algeria.

The Mitidja Plain is a narrow, 100 km long littoral plain located in the Center of Northern Algerian (Figure 1). It extends over an area of about 1450 km$^2$, and it borders the east of the Mediterranean Sea and the north of the Sahel Mountains. To the south, it encloses the Blida Atlas, and to the west, by the mountains of Dahran, it encompasses a latitude of 36° [24]. The soils of this plain are considered the most fertile in Algeria.

This zone is characterized by a Mediterranean climate with a continental influence (sirocco in summer), rainy and mild winters, and hot and dry summers.

The soil is a minimally developed alluvial contribution "fluvisol", with a clay–silt texture, a neutral pH, and no carbonation. It has low content of organic matter, it is well provided with available phosphorus, and it is poor in potassium, which is available according to the Mutcher standards method [25], which corresponds to a content of 0.55 meq/100 g soil for a clay content of about 40% (Table 1). Table 1 provides information on the physical and chemical properties of the soil studied. The data presented in this table are essential for assessing soil fertility and determining the measures to be taken to improve agricultural yields.

**Table 1.** Physical and chemical properties of the studied soil.

| Soil Parameters Analyzed | | The Value |
|---|---|---|
| Physical Traits | Silt (%) | 42.47 |
| | Clay (%) | 40.48 |
| | Potential Hydric | 7.25 |
| | Organic Matter (%) | 1.30 |
| Chemical Properties | CEC (Cmol$^+$ kg$^{-1}$) | 30.09 |
| | Total Nitrogen (mg kg$^{-1}$) | 1710 |
| | Available Nitrogen (mg kg$^{-1}$) | 28.02 |
| | Available Potassium (mg kg$^{-1}$) | 89.7 |
| | Available Phosphorus (mg kg$^{-1}$) | 235.11 |

CEC: Cation Exchange Capacity.

## 2.2. Field Experimental Conditions

### 2.2.1. Climatic Conditions

Figure 2 presents monthly temperature and precipitation data for the experimental site, for two separate agricultural campaigns ("2020–2021" and "2021–2022"), as well as

for the period from 1990 to 2020. The cumulative annual rainfall for the 2020–2021 season was 576.1 mm, and, for 2021–2022, it was 460.3 mm (weather data from the station), corresponding to a normal climatic year. This cumulative annual rainfall between 450–600 mm has been recorded since 2019–2022. During the crop development cycle, 162.8 mm of rain was recorded between December and January. Subsequently, 167.9 mm of rainfall occurred between February and June. The cumulative rainfall for the period of the vegetative cycle of wheat is 330.7 mm in the first year, and it is 320.8 mm in the second year [26].

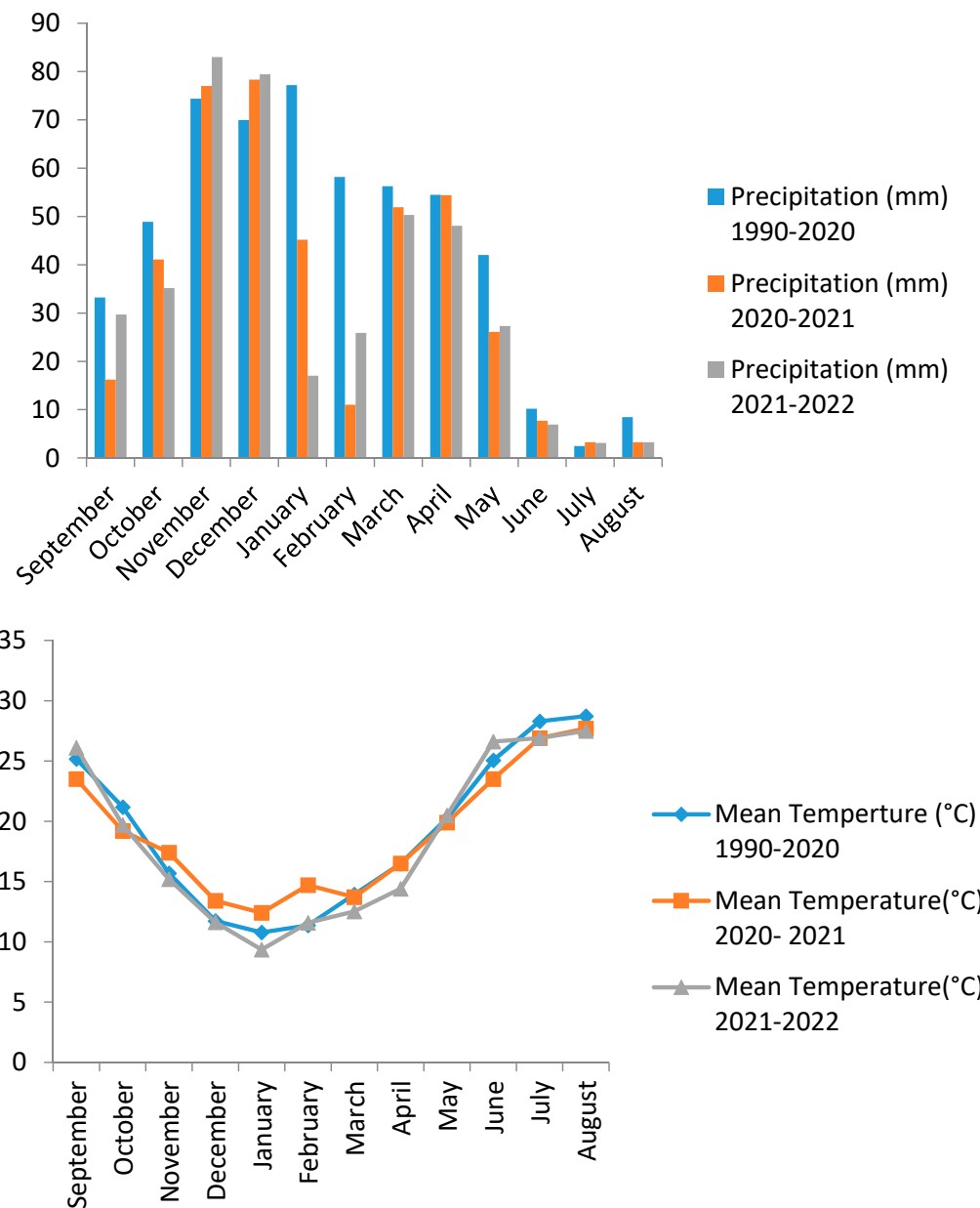

**Figure 2.** Monthly temperatures and precipitation of two agricultural campaigns ("2020–2021"/ "2021–2022") and the period of (1990–2020) for the experimental site.

2.2.2. Experimental Design and Potassium Fertilization

The experiment was carried out according to a Latin square design with five levels of $k_2O$ (K0 = 0 kg $k_2O$ ha$^{-1}$ = Control (no potash fertilizer application), K1 = 100 kg $k_2O$ ha$^{-1}$, K2 = 200 kg $k_2O$ ha$^{-1}$, and K3 = 300 kg $k_2O$ ha$^{-1}$, K4 = 400 kg $k_2O$ ha$^{-1}$), which were repeated in five replicates on 12 m$^2$ elementary plots.

The Berseem was the precedent crop in the field experiment area. The study was carried out with the commonly cultivated variety (cultivar Simeto) of durum wheat. The fertilizer brought before sowing was in the form of potash granulated sulfate (0-0-50 + 18% S). The sowing was performed on 17 December 2020 by a drill in a well prepared seedbed on 18 November 2020 using the seeding rate of 125 kg seed ha$^{-1}$. In the second year, the sowing was performed on 7 January 2022.

Nitrogen and phosphate fertilizers were applied at 69 kg N ha$^{-1}$ in the form of urea and 46 kg $P_2O_5$ ha$^{-1}$ in the form of triple superphosphate. Phosphate fertilizer was applied in a single application before sowing with potassium fertilizer, and nitrogen fertilizer was applied in three different applications (i.e., 23 kg N ha$^{-1}$) at the following stages of development: tillering, first ear, and heading stage.

### 2.3. Soil and Plant Sampling and Measurement

During the crop cycle, dozens of healthy plants were selected from each replicate of each treatment plot, and the following morphological parameters were observed or measured: final vegetation height, spike neck, spike length, and dry matter content at the flowering stage.

At harvest time, dozens of healthy plants were selected from each replicate of each treatment plot. Agronomic observations included: grain yield, number of spikes per square meter, number of grains per spike, and thousand-grain weight, and the protein contents of seed samples were measured by determining the nitrogen content (protein % = nitrogen % $\times$ 6.25) in durum wheat grain [27].

All soil and plant measurements were performed through standard methods. For the soil test, soil samples (dozens of samples) were taken from 0–30 cm depth, before planting. The leaf diagnosis was based on analyzing the second and third leaves of dozens of healthy plants from each replicate of each treatment plot at the flowering stage; mineralization was carried out by an acid attack. N content in both soil and plant samples was determined by the Kjeldahl method [28]. An atomic absorption spectrum using a flame photometer was applied to estimate the potassium concentration of flag leaf and soil samples (potassium extracted with a 1N solution of ammonium acetate). The Olsen method was used for P availability measurement, and organic matter in the soil was measured with the Anne method [29]. All plant and soil measurements were performed within five replicates for each treatment.

### 2.4. Statistical Analysis

Statistical analyses, including the analysis of variance (ANOVA) and means comparison analysis, were carried out using the XLSTAT software, version 2022. The Newman-Keuls test was used to determine significant differences between group means in an analysis of variance at 1% ($p \leq 0.01$), and 5% ($p \leq 0.05$) probability levels was used for means comparison analysis.

## 3. Results
### 3.1. Morphological Characteristics of Durum Wheat

Table 2 shows different morphological parameters that are involved in the vegetative growth of durum wheat. These parameters include plant height, spike length, spike neck, grain number per spike, spike number per m$^2$, and spikelet number per spike. Thus, plant height was significantly affected by the potassium application rate and K-level*season interaction ($p \leq 0.05$). Comparing the two years of experimentation, it was found that the highest plant height was recorded with the 200 kg ha$^{-1}$ k$_2$O rate (85.79 cm for the first year and 83.65 cm for the second year), while the lowest plant height was observed with the 400 kg k$_2$O ha$^{-1}$ application rate (81.73 cm for the first year and 80.93 cm for the second year).

**Table 2.** Effect of potassium levels on morphological traits of durum wheat.

| Season | K-Level | Plant Height (cm) | Spike Length (cm) | Spike Neck (cm) | Grain Number per Spike | Spike Number per m$^2$ | Spikelet Number per Spike |
|---|---|---|---|---|---|---|---|
| | K0 | 83.1 [ab] | 7.08 [a] | 15.94 [b] | 76.97 [a] | 221.9 [b] | 20.68 [a] |
| | K1 | 83.53 [ab] | 7.57 [b] | 18.48 [a] | 79.68 [a] | 226.77 [ab] | 21.61 [a] |
| 2020/2021 | K2 | 85.79 [a] | 7.58 [b] | 17.56 [a] | 79.84 [a] | 220.10 [b] | 22.04 [a] |
| | K3 | 83.46 [ab] | 7.12 [a] | 16.29 [ab] | 77.84 [a] | 213 [b] | 21.12 [a] |
| | K4 | 81.73 [b] | 7.09 [a] | 16.69 [ab] | 78.36 [a] | 219.33 [b] | 20.96 [a] |
| | K0 | 81.95 [b] | 6.57 [a] | 16.06 [b] | 57.27 [b] | 228.25 [ab] | 16.72 [b] |
| | K1 | 82.44 [ab] | 7.04 [b] | 17.51 [a] | 58.96 [b] | 264.25 [a] | 16.97 [b] |
| 2021/2022 | K2 | 83.65 [ab] | 6.76 [ab] | 16.67 [ab] | 61.05 [b] | 251.5 [a] | 17.11 [b] |
| | K3 | 81.95 [b] | 6.61 [a] | 16.19 [ab] | 56.54 [b] | 250 [a] | 16.61 [b] |
| | K4 | 80.93 [b] | 6.58 [a] | 15.98 [b] | 56.53 [b] | 235.51 [ab] | 16.92 [b] |
| *p* values | Cropping Season | 0.15 | 0.09 | 0.23 | 0.01 | 0.02 | 0.02 |
| | K-Level | 0.04 | 0.25 | 0.03 | 0.28 | 0.05 | 0.33 |
| | K-Level*Season Interaction | 0.05 | 0.17 | 0.02 | 0.03 | 0.01 | 0.17 |

Within a column, different letters denote significant differences at $p < 0.05$. k0: 0 kg k$_2$O ha$^{-1}$: Control (No potash fertilizer application); K1: 100 kg k$_2$O ha$^{-1}$; K2: 200 kg k$_2$O ha$^{-1}$; K3: 300 kg k$_2$O ha$^{-1}$; K4: 400 kg k$_2$O ha$^{-1}$.

Spike length was significantly affected by potassium fertilizer levels ($p \leq 0.05$). The highest spike lengths were observed with the 100 kg k$_2$O ha$^{-1}$ and 200 kg k$_2$O ha$^{-1}$ application rates (7.57 cm and 7.58 cm for the first year and 7.04 cm and 6.76 cm for the second year), while the lowest value of spike length was noted with the control plots (7.08 cm for the first year and 6.57 cm for the second year). Therefore, compared to the control, 100 kg k$_2$O ha$^{-1}$ resulted in an increase in spike length of 6.47% and 6.60% for the first year and 7.76% for the second year.

Potassium fertilizer levels and K-level*season significantly influenced the spike neck ($p \leq 0.05$). The 100 kg k$_2$O ha$^{-1}$ rate produced the highest values (18.48 cm for the first year and 17.51 cm for the second year), representing increases of 13.74% and 9.03%, respectively, compared to the control who recorded the lowest values of 15.94 cm and 16.06 cm.

Increasing levels of potassium fertilizer, cropping season, and K-level*season had a significant effect on the spike number per m$^2$ ($p \leq 0.05$). However, the 100 kg k$_2$O ha$^{-1}$ rate produced the highest values for this parameter (226.77 spikes m$^{-2}$ in the first year and 264.25 spikes m$^{-2}$ in the second year). In contrast, the lowest rates were obtained with the 300 kg k$_2$O ha$^{-1}$ rate for the first year (213 spikes m$^{-2}$) and with the control for the second year (228.25 spikes m$^{-2}$).

Finally, morphological parameters of vegetative growth results showed that increasing K fertilizer application had no significant effect on the grain number per spike and the spikelet number per spike ($p > 0.05$). For these two parameters, there was no significant difference between the different levels of K fertilizers tested. These results suggest that other factors, such as irrigation, temperature, and plant variety, may have a greater influence on grain and spikelet production.

### 3.2. Grain Yield and Yield Attributes

#### 3.2.1. Grain Yield

According to the results presented in Table 3, wheat grain yield was significantly affected by the potassium fertilizer application levels ($p \leq 0.05$), the cropping season ($p \leq 0.001$), and the K-level*season ($p \leq 0.01$). The results indicate that the 100 kg k$_2$O ha$^{-1}$ application rate produced the highest grain yields of 2.58 t ha$^{-1}$ in the first year and 2.09 t ha$^{-1}$ in the second year. These numbers represent a significant increase of 23.27% over the control test. However, when the potassium fertilizer rate was increased to 200 kg k$_2$O ha$^{-1}$, a 3.97% decrease in grain yield was observed. Interestingly, the lowest grain yield was obtained with the highest potassium rate (400 kg k$_2$O ha$^{-1}$), i.e., 1.83 t ha$^{-1}$ in the first year and 1.53 t ha$^{-1}$ in the second year. In contrast, using the 100 kg k$_2$O ha$^{-1}$ rate resulted in an increase of 0.54 t ha$^{-1}$ over the control plots.

**Table 3.** Effect of increasing potassium levels on yield traits.

| Season | K-Level | Grain Yield (t ha$^{-1}$) | Dray Matter (%) | Thousand Grain Weight (g) |
|---|---|---|---|---|
| 2020/2021 | K0 | 1.98 [b] | 16.99 [b] | 49.24 [b] |
| | K1 | 2.58 [a] | 18.34 [b] | 53.35 [a] |
| | K2 | 2.45 [a] | 20.01 [a] | 51.2 [a] |
| | K3 | 2.09 [b] | 17.10 [bc] | 48.13 [b] |
| | K4 | 1.83 [ab] | 14.46 [d] | 48.14 [b] |
| 2021/2022 | K0 | 1.55 [c] | 16.18 [c] | 46.8 [bc] |
| | K1 | 2.09 [b] | 16.01 [c] | 49.7 [b] |
| | K2 | 1.9 [ab] | 16.85 [c] | 48.5 [b] |
| | K3 | 1.53 [c] | 16.41 [c] | 45.54 [c] |
| | K4 | 1.67 [b] | 16.35 [c] | 46.74 [bc] |
| *p* Values | Cropping Season | 0.001 | 0.02 | 0.22 |
| | K-Level | 0.03 | 0.05 | 0.04 |
| | K-Level*Season Interaction | 0.01 | 0.03 | 0.01 |

Within a column, different letters denote significant differences at $p < 0.05$. K0: 0 kg $k_2O$ ha$^{-1}$: Control (No potash fertilizer application); K1: 100 kg $k_2O$ ha-1; K2: 200 kg $k_2O$ ha-1; K3: 300 kg $k_2O$ ha-1; K4: 400 kg $k_2O$ ha$^{-1}$.

### 3.2.2. Dry Matter

The results in Table 3 describe the effect of potassium fertilizer levels on wheat crop dry matter. The results indicate that the potassium fertilizer levels, the cropping season, and the K-level*season had a significant impact on dry matter content ($p \leq 0.05$). The use of 200 kg $k_2O$ ha$^{-1}$ of potassium fertilizer resulted in the highest dry matter contents, with 20.01% in the first year and 16.85% in the second year. A dose of 100 kg $k_2O$ ha$^{-1}$ also resulted in a significant increase, with values of 18.34% and 16.5% for the two years, respectively. These results show an increase of 15.09% and 7.36% over the control when both potassium fertilization rates are used. Nevertheless, the use of a 400 kg $k_2O$ ha$^{-1}$ dose generated the least amount of dry matter, with a value of 14.46% for the first year. For the second year, the plots had the lowest amount of dry matter, only 16.35%.

### 3.2.3. Thousand Grain Weight

The data presented in Table 3 highlight the significant effect of potassium fertilizer levels and the K-level*season on the thousand-grain weight of wheat ($p \leq 0.05$; $p \leq 0.01$). The results indicate that application rates of 100 kg $k_2O$ ha$^{-1}$ and 200 kg $k_2O$ ha$^{-1}$ caused a significant increase in this parameter (Table 3). In other words, the use of these two levels of potassium fertilizer may be beneficial in increasing wheat grain weight under the study conditions. The results obtained in the two years of experimentation indicated that the plots that received fertilization at a rate of 100 kg $k_2O$ ha$^{-1}$ produced the maximum thousand-grain weight. Specifically, the thousand grain weight was 53.35 g for the first year and 49.7 g for the second year. In contrast, for plots fertilized with 200 kg $k_2O$ ha$^{-1}$, the thousand-grain weight was 51.2 g in the first year and 48.5 g in the second year. This represents increases of 7.7% and 3.83% over the control plots with the two fertilization rates used. In other words, the application of 100 kg $k_2O$ ha$^{-1}$ appears to be more effective in increasing grain weight than the application of 200 kg $k_2O$ ha$^{-1}$. However, there is a 3.87% decrease in yield when the potassium fertilizer rate is increased from 100 to 200 kg $k_2O$ ha$^{-1}$. Application rates of 300 kg $k_2O$ ha$^{-1}$ and 400 kg $k_2O$ ha$^{-1}$ led to the lowest thousand-grain weights in both years of the experiment (Table 3). In fact, the lowest thousand-grain weight was recorded for these two rates. For the first year, the lowest weights were 48.13 g and 48.14 g for the 300 kg $k_2O$ ha$^{-1}$ and 400 kg $k_2O$ ha$^{-1}$ rates, respectively. For the second year, the lowest weights were 45.54 g and 46.74 g for the same respective doses. These results suggest that the use of high rates of potassium fertilizer may have a negative effect on grain weight.

### 3.3. Protein Content of Wheat Grains

The data presented in Table 4 clearly show that applied K fertilizer levels and K-level*season had a significant effect on the protein amount of wheat grains ($p \leq 0.05$). In general, these results indicate that wheat grain protein content is closely related to potassium fertilizer levels, with an increase in potassium fertilizer rate resulting in an increase in grain protein content.

**Table 4.** Effect of increasing potassium levels on durum wheat protein content.

| Season | K-Level | Protein Content (%) |
|---|---|---|
| 2020/2021 | K0 | 15.30 [ab] |
| | K1 | 16.51 [a] |
| | K2 | 15.82 [ab] |
| | K3 | 15.51 [ab] |
| | K4 | 14.82 [b] |
| 2021/2022 | K0 | 15.08 [b] |
| | K1 | 16.29 [a] |
| | K2 | 15.60 [ab] |
| | K3 | 15.38 [ab] |
| | K4 | 14.88 [b] |
| *p* Values | Cropping Season | 0.42 |
| | K-Level | 0.04 |
| | K-Level*Season Interaction | 0.02 |

Within a column, different letters denote significant differences at $p < 0.05$. k0: 0 kg $k_2O$ ha$^{-1}$: Control (No potash fertilizer application); K1: 100 kg $k_2O$ ha$^{-1}$; K2: 200 kg $k_2O$ ha$^{-1}$; K3: 300 kg $k_2O$ ha$^{-1}$; K4: 400 kg $k_2O$ ha$^{-1}$.

In the first year, protein content fluctuated between 14.82% and 16.51%, while, in the second year, it ranged from 14.88% to 16.29%. The 100 kg $k_2O$ ha$^{-1}$ rate resulted in the highest protein content, 16.51% in the first year and 16.29% in the second year. In contrast, the 200 kg $k_2O$ ha$^{-1}$ rate yielded the second-highest protein content, with 15.82% in the first year and 15.6% in the second year, respectively. On the other hand, the results of the study revealed that the application rate of 400 kg $k_2O$ ha$^{-1}$ led to the lowest protein content of 14.82% in the first year and 14.88% in the second year. These numbers indicate that excess $k_2O$ can have a negative impact on protein production in the crop. Therefore, finding the right balance in terms of $k_2O$ application rate is essential to maximize protein production. These results can help farmers adjust their fertilizer use to optimize crop quality and yield.

Based on these results, we observed a significant increase in protein content with potassium fertilizer application compared to control plots. Specifically, we found increases of 7.33%, 3.29%, and 1.35% at 100 kg $k_2O$ ha$^{-1}$, 200 kg $k_2O$ ha$^{-1}$, and 300 kg $k_2O$ ha$^{-1}$, respectively. However, it should be noted that potassium fertilizer application resulted in a decrease in crop protein content. Indeed, in the second year, we observed an increase of 7.43% compared to the control plots with a rate of 100 kg $k_2O$ ha$^{-1}$ (Table 4).

### 3.4. Nutritional Status of Plants

The data presented in Table 5 indicate that the application of potassium fertilizer levels of 300 kg $k_2O$ ha$^{-1}$ and 400 kg $k_2O$ ha$^{-1}$ resulted in an increase in the plant potassium content, reaching 2.03 g kg$^{-1}$ for both doses in the first year, as well as 1.98 g kg$^{-1}$ with the 300 kg $k_2O$ ha$^{-1}$ dose and 2.01 g kg$^{-1}$ with the 400 kg $k_2O$ ha$^{-1}$ dose for the second year, respectively. However, it should be emphasized that, for both years of experimentation, the results did not show a significant effect for the cropping season factor studied ($p > 0.05$). On the other hand, for the other two factors, namely, the K-level and K-level*season interaction, the differences observed in plant potassium content were statistically significant ($p \leq 0.05$). The results also revealed that the control plots had the lowest leaf potassium levels, with minimum values of 1.84 g kg$^{-1}$ for the first year and 1.67 g kg$^{-1}$ for the second year, respectively.

**Table 5.** Potassium and nitrogen contents in wheat leaves at flowering stage.

| Season | K-Level | K (g kg$^{-1}$) | N (%) |
|---|---|---|---|
| 2020/2021 | K0 | 1.84 [ab] | 2.95 [ab] |
| | K1 | 1.91 [a] | 2.8 [a] |
| | K2 | 1.94 [a] | 2.83 [a] |
| | K3 | 2.03 [a] | 2.98 [b] |
| | K4 | 2.03 [a] | 2.87 [ab] |
| 2021/2022 | K0 | 1.67 [b] | 2.91 [ab] |
| | K1 | 1.73 [ab] | 2.84 [a] |
| | K2 | 1.77 [ab] | 2.87 [a] |
| | K3 | 1.98 [a] | 2.96 [b] |
| | K4 | 2.01 [a] | 2.91 [ab] |
| *p* values | Cropping Season | 0.15 | 0.51 |
| | K-Level | 0.05 | 0.27 |
| | K-Level*Season Interaction | 0.03 | 0.55 |

Within a column, different letters denote significant differences at $p < 0.05$. K%: Potassium, N%: Nitrogen, k0: 0 kg $k_2O$ ha$^{-1}$: Control (No potash fertilizer application); K1: 100 kg $k_2O$ ha$^{-1}$; K2: 200 kg $k_2O$ ha$^{-1}$; K3: 300 kg $k_2O$ ha$^{-1}$; K4: 400 kg $k_2O$ ha$^{-1}$.

In addition, analyses were conducted to assess the nitrogen content of the leaves of the wheat crop, and the results are summarized in Table 5. This assessment provides a better understanding of the nutritional status of the plant in terms of nitrogen, which is a crucial element for the growth and development of the crop. It was found that the plots that received 300 kg $k_2O$ ha$^{-1}$ fertilization had the highest nitrogen contents, with 2.98% in the first year and 2.96% in the second year. However, an application of 100 kg $k_2O$ ha$^{-1}$ led to the lowest N contents, at 2.81% in the first year and 2.84% in the second year.

In general, the analyses showed that crop leaf N contents were in the range of 2.81% to 2.96% for both years of experimentation and that there was no significant correlation with the levels of potash fertilizer applied. In other words, the application of potassium fertilizer at different levels did not have a significant impact on leaf nitrogen levels in the study crop ($p > 0.05$). It is important to note that leaf nitrogen content is a key indicator of plant nitrogen nutrition, which can have a significant impact on growth, yield, and crop quality. The results of this analysis suggest that, in this particular context, leaf nitrogen content does not appear to be influenced by the levels of potassium fertilizer applied.

*3.5. Residual Soil Potassium*

The data in Table 6 indicate that the average amount of potassium available in the soil varied with the cropping season, the rates of potassium fertilizer applied, as well as the K-level*season interaction. The results also show that the amount of potassium available in the soil after harvest was significantly influenced by potassium fertilizer application levels in both years of experimentation ($p \leq 0.05$). This trend is also observed for the cropping season and K-level*season interaction, but, in this case, the differences are highly and very highly significant, respectively ($p \leq 0.01$; $p \leq 0.001$). Potassium fertilizer rates influenced the levels of available potassium in the soil. The highest available potassium levels were obtained with the 400 kg $k_2O$ ha$^{-1}$ application level, with values of 144.3 mg kg$^{-1}$ for the first year and 109.22 mg kg$^{-1}$ for the second year. In contrast, the lowest levels were recorded in the control plots, reaching 81.9 mg kg$^{-1}$ for the first year and 74.1 mg kg$^{-1}$ for the second year. It is important to emphasize that the application of 300 kg $k_2O$ ha$^{-1}$ also resulted in a concentration equivalent to that measured in the control plots, but only during the first year of the experiment. These results clearly indicate the importance of applying potassium fertilizer to increase the availability of potassium in the soil, which can have positive effects on the growth and production of the crops grown there.

**Table 6.** Post-harvest available potassium content (meq/100 g soil).

| Season | K-Level | Average K Available (mg kg$^{-1}$) |
|---|---|---|
| 2020/2021 | K0 | 81.9 [cd] |
| | K1 | 89.7 [c] |
| | K2 | 85.8 [c] |
| | K3 | 81.9 [a] |
| | K4 | 144.3 [a] |
| 2021/2022 | K0 | 74.1 [d] |
| | K1 | 78.01 [cd] |
| | K2 | 78.39 [cd] |
| | K3 | 85.85 [c] |
| | K4 | 109.22 [b] |
| *p* Values | Cropping Season | 0.01 |
| | K-Level | 0.02 |
| | K-Level*Season Interaction | 0.001 |

Within a column, different letters denote significant differences at $p < 0.05$. k0: 0 kg $k_2O$ ha$^{-1}$: Control (No potash fertilizer application); K1: 100 kg $k_2O$ ha$^{-1}$; K2: 200 kg $k_2O$ ha$^{-1}$; K3: 300 kg $k_2O$ ha$^{-1}$; K4: 400 kg $k_2O$ ha$^{-1}$.

### 3.6. Yield and its Correlation Analyses with Spike Length and Thousand Grain Weight

In this study, data were collected on two agronomic characteristics of durum wheat, namely, spike length and thousand-grain weight. Then, these data were analyzed to assess their correlation with grain yield. For this purpose, simple linear regression was used, a statistical method commonly used to evaluate the relationship between two variables. Figure 3 shows the results of this analysis, indicating the relationship between two previously mentioned agronomic characteristics and grain yield. The results showed that grain yield was strongly correlated with thousand-grain weight ($r^2 = 0.91$), as well as with spike length ($r^2 = 0.90$ ***). Thus, this study determined whether ear length and thousand-grain weight were reliable indicators of grain yield, which may be useful for selecting the best-performing wheat varieties and optimizing farming practices to increase grain yield.

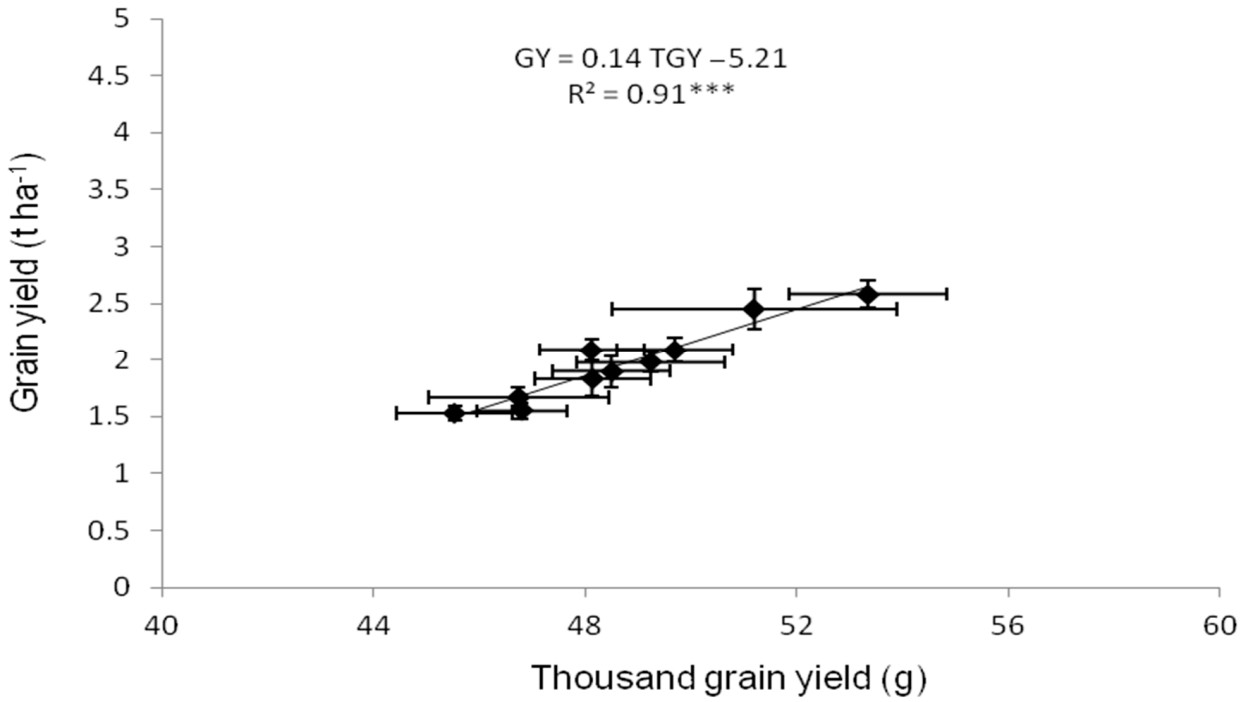

**Figure 3.** *Cont.*

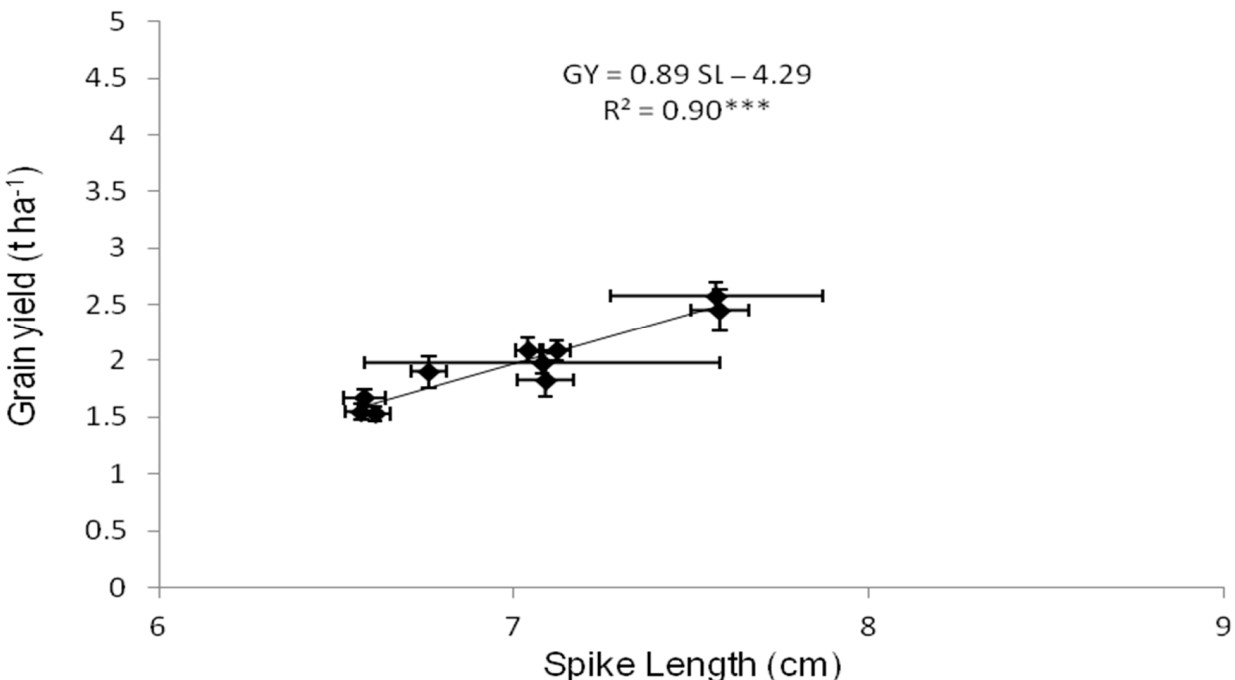

**Figure 3.** Regression equations for the relationship of grain yield (GY) to spike length (SL) and thousand-grain weight (TGY). Asterisk "***" denotes significant difference $p \leq 0.001$, respectively. All plotted data are the mean of five replicates for each measured variable. Linear regression was established between GY and both SL and TGY mean values measured during the two cropping seasons and under each K-level treatment. The bars represent standard errors from five replicate of either GY (vertical bare) or SL and TGY (horizontal bar) mean values.

## 4. Discussion

From the parameters studied in this study, it was found that the application of a potassium dose of 100 kg $k_2O$ ha$^{-1}$ results in the highest grain yield and protein content (Tables 3 and 4). Indeed, this content led to a yield increase of 600 kg ha$^{-1}$, which corresponds to a productivity index of 6 kg of grain per kg of $k_2O$. In Mediterranean climate regions, the practice of potassium fertilization is essential to maximize the grain yield of a crop. However, its effectiveness depends on plant water conditions.

It is also important to remember that the soil of the experimental plot lacks available potassium and that, even at harvest, its available potassium content does not meet Mutcher's standard, which recommends an optimal level of 0.55 meq/100 g of soil [25]. Potassium fertilizer levels used are insufficient to achieve optimum potassium fertility, mainly due to crop removal and the potassium retrogradation phenomenon in the soil [26].

Moreover, it should also be noted that the climatic conditions of the region during the two years of experimentation were favorable and that the total amount of rainfall for the wheat vegetative cycle was only 330.7 mm in the first year and 320.8 mm in the second year, which is significantly insufficient compared to the typical requirements of a cereal crop, estimated at around 450 mm throughout its growth cycle [26]. The lack of water during the growth phase resulted in an unbalanced absorption of vital nutrients, which prevented the plant from growing and developing normally, thus resulting in a limited height. The negative effect of K fertilizers on plant height was confirmed by this result, which is consistent with the finding of Hefni [30]. The study results show that grain number per spike, and spikelet number per spike, did not show any significant difference under the different potassium fertilizer rates used. However, spike length, plant height, spike neck, and spike number per m$^2$ displayed statistically significant differences with the influence of the three doses of potassium. It is important to note that these results diverge significantly from those obtained in the previous study conducted by Maurya [31]

for parameters that did not show significant differences. At the same time, the results are consistent for parameters that showed significant differences.

The research conducted by Lateef and Altamimi [32] has shown that potassium fertilization can improve agronomic parameters, despite water constraints. According to these studies, the application of potassium fertilizer at doses of 100 kg $k_2O$ $ha^{-1}$ and 200 kg $k_2O$ $ha^{-1}$ can optimize certain agronomic parameters, such as spike length (200 kg $k_2O$ $ha^{-1}$) and spike neck (100 kg $k_2O$ $ha^{-1}$). These results are encouraging for farmers who have to face difficult weather conditions while seeking to improve their production.

According to the works of Samar Raza et al. [33], as well as Faisal Mihbes Al-Taher and Howraa Hussein AL-Naser [34], the application of potassium fertilization can also improve agricultural yields. However, the optimal amount of fertilization varies depending on yield parameters. For dry matter content, the highest value is obtained with a dose of 200 kg $k_2O$ $ha^{-1}$, while, for other characteristics, such as 1000 grain weight, grain yield, and protein content, the optimal dose is 100 kg $ha^{-1}$ $k_2O$.

The research results have demonstrated that potassium plays an important role in water use efficiency by plants, improving their growth and cell division, as well as the production of proteins and hydrocarbons that are rapidly transferred to the grains [1]. Potassium fertilizer levels also had a significant effect on thousand-grain weight ($p < 0.05$), which is consistent with the findings of Ghulam et al. [35]. However, the highest grain yield was obtained with a rate of 100 kg $k_2O$ $ha^{-1}$ during the two campaigns, which seems to be relatively low compared to years when weather conditions are favorable [36]. The crop yield was increased by 0.6 t $ha^{-1}$ compared to the control trial in the first year and by 0.54 t $ha^{-1}$ in the second year. These results are in agreement with the findings of Mirza et al. [37], who confirm that potassium is directly or indirectly responsible for this increase in yield by regulating the biosynthesis, conversion, and allocation of metabolites. Moreover, these results are consistent with the findings of Ghulam et al. [35], as well as Mohamad Javad Raisi and Enayatollah Tohidi-Nejad [38].

The application of various levels of potassium fertilizer had a significant impact on grain protein content ($p \leq 0.05$), which is a key quality parameter of the studied wheat. This result is in agreement with the finding reported by Alam et al. [39].

For optimal nutrition, it seems that the application of 100 kg $K_2O$ $ha^{-1}$ is sufficient to reach adequate potassium levels with foliar $K_2O$ content $\geq 2.30\%$ [38]. Similarly, this supply also appears to be sufficient to achieve adequate nitrogen levels in the leaves with $K_2O \geq 2.50\%$ [40]. According to the results of this present study, we suggest that the potassium content in the soil would be sufficient to meet the nutritional needs of the plant, which would promote healthy and optimal growth. However, it should be noted that potassium requirements can vary depending on various factors, such as crop type, environmental conditions, and soil management practices. It is, therefore, important to assess the potassium requirements of each specific crop before determining the optimal application rate.

The regression analysis results have shown a positive correlation between grain yield, spike length, and thousand-grain weight. These plant characteristics are, therefore, considered key factors for improving yields and selecting wheat genotypes better adapted to low potassium and water deficit conditions. In conclusion, it can be said that grain yield has improved throughout wheat development. The results obtained by Qiang Lu et al. [41] showed similar positive correlations for wheat, indicating that the increase in grain yield following K fertilizer treatments was mainly due to an increase in thousand-grain weight. On the other hand, numerous studies have also noted a positive correlation between grain yield and ear length [42–44], which is consistent with our own results.

The key results of this study suggest that the reasoned application of potassium fertilizer on durum wheat cultivars can improve the growth of morphological traits and yield quality. However, this improvement depends on plant water and seasonal weather conditions. It was observed that a potassium fertilizer rate of 100 kg $k_2O$ $ha^{-1}$ is sufficient to obtain satisfactory grain yield under the study conditions. It should be noted that the

optimum amount of potassium fertilizer may vary depending on environmental conditions and soil properties. Thus, a higher rate than recommended may not lead to a significant increase in yield and may even result in nutrient loss due to excessive leaching. In addition, the plant may not assimilate nutrients properly, which could hinder its growth and development, resulting in low yields. Therefore, it is important to adjust the amount of potassium fertilizer, according to the needs of each plot of land, to optimize crop production. Furthermore, the application of potassium fertilizer should not be considered the only solution to improve yields. Other cultural practices, such as water management, crop rotation, plant protection, and the use of other types of fertilizers, can also have a significant impact on durum wheat production. Therefore, it is important to consider these factors in a comprehensive approach to maximize yields and maintain soil health.

## 5. Conclusions

Although the applied potassium rates are adequate to meet the nutrient requirements of durum wheat, they are not sufficient to improve the potassium fertility of the plot soil. Thus, in order to maintain optimal wheat crop productivity and soil fertility in the long term, it is essential to compensate for potassium losses caused by crops and climatic conditions, such as water deficit, by applying balanced and adequate potassium fertilization. Despite the water scarcity during the study period, the application of potassium fertilization resulted in a significant improvement in production, both in terms of quantity (grain yield) and quality (grain quality, measured through protein content). These results can be extrapolated to other soil types in this region and to other wheat genotypes in future experiments.

**Author Contributions:** Conceptualization, M.L. and N.Y.R.; methodology, M.L. and F.L.; software, A.M.; validation, O.A.Z. and N.Y.R.; formal analysis, M.D. and S.H.; investigation, A.M. and M.L.; resources, M.D. and N.Y.R.; data curation, S.H.; writing—original draft preparation, N.Y.R. and M.L.; writing—review and editing, D.E.K. and W.O.; visualization, M.L.; supervision, F.L. and O.A.Z.; project administration, D.E.K. All authors have read and agreed to the published version of the manuscript.

**Funding:** This publication has been supported by the RUDN University Scientific Projects Grant System, project No. <202724-2-000>.

**Acknowledgments:** Special thanks to the National Higher School of Agriculture and the Technical Institute of Extensive Cultivation in Oued Semar Algiers for their precious help in the realization of this study. Authors thank also Alliouche Ahmed for his help in producing the map of case study.

**Conflicts of Interest:** The authors declare no conflict of interest.

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
