# Peer review of "Investigating the Potassium Fertilization Effect on Morphological and Agrophysiological Indicators of Durum Wheat under Mediterranean Rain-Fed Conditions"

_agriculture, doi:10.3390/agriculture13061142_

Round 1

Reviewer 1 Report

I have evaluated this manuscript (agriculture-2330013) entitled “Investigating the Effect of Potassium Application on Growth and Grain Yield of Durum Wheat underRainfed Mediterranean conditions” submitted for publication in ‘Agriculture’. Topic of manuscript is interesting and falls within the scope of journal. However, in its current form, the manuscript has several serious issues given below.

1. I have serious concerns about the novelty of this work. This manuscript only studied the effect of potassium application on morphological traits and grain yield of durum wheat, there was no innovation in research contents and methods.

2. The “Climatic Conditions” description of the Materials and Methods section should be changed to a Figure. Meanwhile, “This rainfall is lower than the usual needs of a cereal, which would be 450 mm for the duration of the crop cycle [24].” is redundant here. Moreover, “2.2.2. Experimental Design” and “2.2.3. Potassium fertilization” should be combined. The description in “Soil and plant sampling and measurement” is missing important information. What is the number of repetitions of the experiment? How are the plants and soil determined?

3. Why are some characters in the Results section not significantly analyzed, such as “Spike Length” and “N%”?

4. The key words in the manuscript are summarized inappropriately.

5. The serial numbers of Figures and Tables in the manuscript are confusing.

 Extensive editing of English language required

Author Response

Dear Reviewer 1 :

I would like to thank you very much for your constructive comments on our research, which contribute to its improvement.

Response

  1. I have serious concerns about the novelty of this work. This manuscript only studied the effect of potassium application on morphological traits and grain yield of durum wheat, there was no innovation in research contents and methods.

Response 1: First, we inform you dear reviewer that during this revision we mentionned and exlained the novetly of our work in both abstract, discussion and conclusion section. In addition, we inform you that the present work have two points of novetly at both national and international scale, as that :

  • Regarding of the old and recent literature: we have little work that were performed within the proposition of our dose K-level that we applied here in our work. Our proposed methodology in terms of the choice of the different levels of fertilization are proposed in the first time in this work. Indeed, our field experiment was performed under organic farming system without addition of any fertilizer except to K, this led to better study the effect of K and which with no possible interaction with another fertilizer nutrient.
  • At national scale, the present work is considered as one of the few work that assess the K-fertilization effect in sub-humid Algerian area where durum wheat is cultivated. Indeed, our research is considered as the first work that assess the effect of K fertilization on both grain yield and yield quality with the assessment of all yield compounent.

  1. The “Climatic Conditions” description of the Materials and Methods section should be changed to a Figure. Meanwhile, “This rainfall is lower than the usual needs of a cereal, which would be 450 mm for the duration of the crop cycle [24].” is redundant here. Moreover, “2.2.2. Experimental Design” and “2.2.3. Potassium fertilization” should be combined. The description in “Soil and plant sampling and measurement” is missing important information. What is the number of repetitions of the experiment? How are the plants and soil determined?

Response 2:

- The climatic data has been put in graphic form upon your request (plz see Fig 2).

-The sentence “This rainfall is lower than the usual needs of a cereal, which would be 450 mm for the duration of the crop cycle“, This statement was made by the FAO based on its research into the water requirements of durum crops during their development and growth cycle  has been removed

-  The section of “2.2.2. Experimental Design” and “2.2.3. Potassium fertilization” have been combined (plz see paragraph L : 129-143).

- What is the number of repetitions of the experiment ? we have mentioned in the lines : 130-132 , that the experiment was carried out according to a Latin square design with five levels of k2O (K0 = control (no potash fertilizer application), K1 = 100 kg K2O ha -1, K2 = 200 kg K2O ha -1, K3 = 300 kg K2O ha -1, K4 = 400 kg K2O ha-1) repeated in 5 replicates on 12 m2 elementary plots.

 -How are the plants and soil determined? Plz see paragraphs (145-155).

  1. Why are some characters in the Results section not significantly analyzed, such as “Spike Length” and “N%”?

Response 3: it was mistakenly forgotten due to the large amount of data, and the results of its analysis were mentioned after your constructive comment. (plz see Table 2 and 5).

  1. The key words in the manuscript are summarized inappropriately.

Response 4: we work to improve the key words of our work appropriately.

  1. The serial numbers of Figures and Tables in the manuscript are confusing.

Response 5: the serial numbers of Figures and Tables in the manuscript have been corrected.

Reviewer 2 Report

The paper is well written however doesn't have the novelty. Only different doses of potassium application on wheat yield is not enough to be published. I don't see any novelty in the work. The author's must understand the significance of their works. You can find 100a of papers on NPK or one these applications effect on crop yield.

Author Response

Dear Reviewer 2 :

I would like to thank you very much for for your contribution to evaluate our work. In response to  your comment :

The paper is well written however doesn't have the novelty. Only different doses of potassium application on wheat yield is not enough to be published. I don't see any novelty in the work. The author's must understand the significance of their works. You can find 100a of papers on NPK or one these applications effect on crop yield.

Response: First, we inform you dear reviewer that during this revision we mentionned and exlained the novetly of our work in both abstract, discussion and conclusion section. In addition, we inform you that the present work have two points of novetly at both national and international scale, as that :

  • Regarding of the old and recent literature: we have little work that were performed within the proposition of our dose K-level that we applied here in our work. Our proposed methodology in terms of the choice of the different levels of fertilization are proposed in the first time in this work. Indeed, our field experiment was performed under organic farming system without addition of any fertilizer except to K, this led to better study the effect of K and which with no possible interaction with another fertilizer nutrient.
  • At national scale, the present work is considered as one of the few work that assess the K-fertilization effect in sub-humid Algerian area where durum wheat is cultivated. Indeed, our research is considered as the first work that assess the effect of K fertilization on both grain yield and yield quality with the assessment of all yield compounent.

Reviewer 3 Report

1) The title of the manuscript 'Investigating the Effect of Potassium Application on Growth and Grain Yield of Durum Wheat under Rainfed Mediterranean Conditions' should be changed as it does not entirely correspond to the subject matter contained in it. The paper does not present the results of research on the effect of K fertilization on the course of plant vegetation.

2) Yield amount - should be given according to SI (t∙ha-1), not in q∙ha-1

3) The notation "100-km long" should be corrected (L: 96)

4) In Figure 2, "The extent of the Mitidja plain in northern Algeria." it is worth indicating the location of the experiment.

5) In my opinion, the title of subsection 3.1 "Morphological Parameters of Vegetative Growth" (L 164) is not appropriate and should be changed. In it, the authors present morphological features of durum wheat.

6) Table 1 shows plant morphological features that are not involved in the vegetative growth of durum wheat (so the authors claim), but are the result of the process of plant growth and development (L: 165-166)

7) Please correct the notation "k2O" (L: 123), "k2O ha-1" (e.g., L: 186, 189, 199 and further in the manuscript)

8) "granulated sulfate of potash" (L: 128) - please specify the potassium content of the fertilizer.

9) "Olsen method was used for P availability measurement, and organic matter in the 155 soil was measured with the Anne method." (L: 155-156)-please indicate the literature for these analytical methods.

10) L: 288-293 - I suggest moving this piece of text to the discussion.

11) In my opinion, the determination of the regression equation for 5 data is not appropriate (too small a sample size). In addition, Figure 1. shows regression equations for the relationship of grain yield to: a) spike length, b) thousand-grain weight, and not as stated "Regression analysis results between spike length, thousand-grain weight, and grain yield." (L: 348)

12) Discussion - I suggest an attempt to explain why higher doses of potassium in the cultivation of Triticum durum, above 100 kg K∙ha-1, resulted in a decrease in yield and other studied characteristics.

13) References - adapt the literature citation to the requirements of the journal Agriculture

Author Response

Dear Reviewer 3 :

I would like to thank you very much for your constructive comments on our research, which contribute to its improvement.

1) The title of the manuscript 'Investigating the Effect of Potassium Application on Growth and Grain Yield of Durum Wheat under Rainfed Mediterranean Conditions' should be changed as it does not entirely correspond to the subject matter contained in it. The paper does not present the results of research on the effect of K fertilization on the course of plant vegetation.

Response 1: we are working to improve the title of the research according to their objectives.

2) Yield amount - should be given according to SI (t∙ha-1), not in q∙ha-1

Response 2: the units have been corrected.

3) The notation "100-km long" should be corrected (L: 96)

Response 3: the notation was been corrected. (plz see L :97).

4) In Figure 2, "The extent of the Mitidja plain in northern Algeria." it is worth indicating the location of the experiment.

Response 4: we have indicated the location of the experiment (plz see L :92) according to Figure 2.

5) In my opinion, the title of subsection 3.1 "Morphological Parameters of Vegetative Growth" (L 164) is not appropriate and should be changed. In it, the authors present morphological features of durum wheat.

Response 5: the title of "subsection 3.1"  has been changed to : Morphological Parameters of plants.

6) Table 1 shows plant morphological features that are not involved in the vegetative growth of durum wheat (so the authors claim), but are the result of the process of plant growth and development (L: 165-166)

Response 6: Done. The table title was rephrased

7) Please correct the notation "k2O" (L: 123), "k2O ha-1" (e.g., L: 186, 189, 199 and further in the manuscript)

Response 7 : Throughout the manuscript, corrections have been made to the "K2O" notation.

8) "granulated sulfate of potash" (L: 128) - please specify the potassium content of the fertilizer

Response 8: the granulated sulfate of potash used is : 0-0-50+18%S ( plz see L :135).

9) "Olsen method was used for P availability measurement, and organic matter in the 155 soil was measured with the Anne method." (L: 155-156)-please indicate the literature for these analytical methods.

Response 9: the literature for these analysis methods has been carried out added (plz see L : 163-164).

10) L: 288-293 - I suggest moving this piece of text to the discussion

Response 10: this piece has been transferred to the discussion section.

11) In my opinion, the determination of the regression equation for 5 data is not appropriate (too small a sample size). In addition, Figure 1. shows regression equations for the relationship of grain yield to: a) spike length, b) thousand-grain weight, and not as stated "Regression analysis results between spike length, thousand-grain weight, and grain yield." (L: 348)

Response 11: determination of the regression equation for 5 data, and  with five repetitions, thus a total of 25 data for each parameter.

Figure 3 : shows regression equations for the relationship of grain yield to: a) spike length, b) thousand-grain weight

12) Discussion - I suggest an attempt to explain why higher doses of potassium in the cultivation of Triticum durum, above 100 kg K∙ha-1, resulted in a decrease in yield and other studied characteristics.

Response 12: short explanation

 The reason why most of the characteristics decrease with the increase of the amount of potassium above 100 kg K ha-1 is probably that in the presence of amounts higher than this, we will have a poor assimilation of the elements by the plant and therefore a malfunctioning of the plants, causing a decrease in growth and a bad development of the crop.

 13) References - adapt the literature citation to the requirements of the journal Agriculture

Response 13: we have corrected some references, as required by the journal Agriculture.

Reviewer 4 Report

Paper has been nicely written and can be accepted after minor revision. points are given below

1- update the references, to give the current state of knowledge

2- update the hypothesis and objectives. these are confusing 

3- conclusion section is too lengthy. give a short and concert conclusion. do not repeat the results in this section.  

It is OK

Author Response

Comment: Paper has been nicely written and can be accepted after minor revision. points are given below

1- update the references, to give the current state of knowledge

Response 1: We update a new and recent reference (2019, 2021, 2022...etc) that are considered as potential references that were performed in our same context

2- update the hypothesis and objectives. these are confusing 

ResponseDone. please see the new introduction section

3- conclusion section is too lengthy. give a short and concert conclusion. do not repeat the results in this section.  

Response: conclusion was rephrased and within less word as your recommendation

Round 2

Reviewer 2 Report

Different levels of major elements doesn't reflect any novelty in the field of research. For instance, a paper published with xyz levels of NPK respectively, now would it be novelty if I change the levels of NPK? For me it's not. Yes, it is if accompaned with different technologies or amendments, again for example; different levels of K under organic amendment could be be a novelty ? Do you have such direction, please indicate? 

 The author should think broadly and globally, even the author mentioned that on national scale it is one of the few new work, not the only, but it should be globally one of the few.  

I absolutely stands on my last review. 

Author Response

Response to reviewer 1

1)Dear reviewer I inform you that our approch was based on the K-fertilization under organic farming practices that are commonly applied by cereals farmers under rainfed conditions of sub-humid regios in Algeria. So, in terms of organic fertilization,  In our research we precisely based on K-Mineral fertilizer and we prefer to focus on to better stabilize the cycling of this element between soil-plant organ and not give the opportunity to other fertilizer that could lead to make more interaction with K that is present in both soil and uptake by plant. 2) For applied K-levels, yes we proposed a novel rate of each applied dose and this was performed after lob literature research and discussion with farmers, So, according to our major finding this lead to propose our results to better optimize the use of K-fertilizer by farmers to better reduce K-fertilizer loss. So this finding is considered as a novel as compared to both N and P fertilizers application and this is also considered as the first one research that present this approach and also finding at national scale and as one of the most few finding in international scale under our experiment condition (soil, climate and cereals variety). 3) For technology application, we proposed our scientific approach after considering ( literature and previous research) the most applied soil management technical and that which was commonly applied by farmers, so we want to test our hypothesis under the common practices that are not target by the objective of our study (i.e. soil management,  organic  fertilization, planting density…..etc). So our directives are mainly focused in how to optimize the K-fertilization in durum wheat crop to better improve and control growth and yield component under Mediterranean rainfed conditions. In addition, I inform you that our case study is considered as typical region of Mediterranean area in particular as compared to the Mediterranean countries that chair an approximate soil-climate condition.

All of these informations were mentioned now in the text in the end of the manuscript. As that the added correction is : “Despite the importance of the role of potassium in the formation of yield, the practice of potassium fertilization on the wheat crop remains very limited in Algeria [23]. The main objective of the present study is to identify the optimal K-fertilizer dose that could stabilize and improve growth and yield components of durum wheat crop, particularly under organic cropping management in sub-humid climate. We hypothesize that both low and moderate K-application will maintain simultaneously yield and K-acquisition by durum wheat. The specific objectives of this study were (i) to evaluate the influence of different K fertilizer application rates on wheat yield, (ii) to understand the response of the crop to variation in K use on morphological traits with different rates of potash fertilizer, and (iii); to determine the influence of potassium rates on the quality of wheat grain. The field experiment was conducted within the same conditions of crop and soil managements that are commonly practiced by the local farmers under rain-fed conditions.

Reviewer 3 Report

Dear Author,

The Authors have made the revisions to their work indicated in the review. I still have doubts about the correctness of the statistical calculations. I stand by my opinion that determining the regression equation for 5 data is not appropriate (too small a sample size). The more observations we have, the more accurate our estimate is by which the confidence interval will be narrower. I suggest calculating regression equations for all the data of each parameter - in response to comments in the review, the Authors indicate that there are 25 data for each parameter.

Author Response

Response to reviewer 2

Done. Please see the new figure 3. Dear reviewer, we agree perfectly with your comment. So as your recommendation we improved globally the figure 3 by adding these informations:

  • We plotted all measured values of grain yield with spick length and the weight of 1000 grain during the two cropping season. So, Linear correlation was established between all WEU and NUE values measured during the four cropping seasons and under each N-level traitment. Within 4 replicate for each N-level*year traitment.
  • For replication we prefere to represent replication (5) by standar error in the regression to better indicate the precision of measurement at the mean values.
